# Antibody Generation and Immunogenicity Analysis of EBV gp42 N-Terminal Region

**DOI:** 10.3390/v13122380

**Published:** 2021-11-28

**Authors:** Junping Hong, Dongmei Wei, Qian Wu, Ling Zhong, Kaiyun Chen, Yang Huang, Wanlin Zhang, Junyu Chen, Ningshao Xia, Xiao Zhang, Yixin Chen

**Affiliations:** 1National Institute of Diagnostics and Vaccine Development in Infectious Diseases, School of Life Sciences, Xiamen 361005, China; hongjunping1992@outlook.com (J.H.); weidongmei@stu.xmu.edu.cn (D.W.); wuqiann@stu.xmu.edu.cn (Q.W.); chenkaiyun@stu.xmu.edu.cn (K.C.); leolovege@stu.xmu.edu.cn (Y.H.); JunyuChen@xmu.edu.cn (J.C.); nsxia@xmu.edu.cn (N.X.); 2State Key Laboratory of Molecular Vaccinology and Molecular Diagnostics, School of Public Health, Xiamen University, Xiamen 361005, China; 3State Key Laboratory of Oncology in South China, Collaborative Innovation Center for Cancer Medicine, Guangdong Key Laboratory of Nasopharyngeal Carcinoma Diagnosis and Therapy, Sun Yat-sen University Cancer Center, Guangzhou 510060, China; zhongling@sysucc.org.cn (L.Z.); zhangwl2@sysucc.org.cn (W.Z.)

**Keywords:** Epstein–Barr virus (EBV), glycoprotein 42, gp42 N-terminal region, rhesus lymphocryptovirus (rhLCV), virus-like particles (VLPs)

## Abstract

Epstein–Barr virus (EBV) is the first reported oncogenic virus and infects more than 90% of adults worldwide. EBV can establish a latent infection in B lymphocytes which is essential for persistence and transmission. Glycoprotein gp42 is an indispensable member of the triggering complex for EBV entry into a B cell. The N-terminal region of gp42 plays a key role in binding to gH/gL and triggering subsequent membrane fusion. However, no antibody has been reported to recognize this region and the immunogenicity of gp42 N-domain remains unknown. In the present study, we have generated a panel of nine mAbs against the gp42 N-terminal region (six mAbs to gp42-44-61aa and three mAbs to gp42-67-81aa). These mAbs show excellent binding activity and recognize different key residues locating on the gp42 N-domain. Among the nine mAbs, 4H7, 4H8 and 11G10 cross-react with rhLCV-gp42 while other mAbs specifically recognize EBV-gp42. Our newly obtained mAbs provide a useful tool for investigating the gp42 function and viral infection mechanism of γ-Herpesvirus. Furthermore, we assess the immunogenicity of the gp42 N-terminal region using the HBc149 particle as a carrier protein. The chimeric VLPs can induce high antibody titers and elicit neutralizing humoral responses to block EBV infection. More rational and effective designs are required to promote the gp42-N terminal region to become an epitope-based vaccine.

## 1. Introduction

Epstein–Barr virus (EBV), also called Human Herpesvirus 4 (HHV-4), is a member of the human herpesvirus family and belongs to the gammaherpesvirus sub-family [1,2]. Like other herpesviruses, EBV infections are ubiquitous and EBV establishes long-term latency in infected human hosts [3]. As the first identified human oncogenic virus, EBV is regarded as a co-factor in the development of several malignancies including lymphoid malignancies and nasopharyngeal carcinoma [4]. EBV mainly infects two target cells, epithelial cells and B cells, which is a complex process mediated by multiple viral envelope glycoproteins [5]. gH, gL and gB are needed for the infection of both cell types [6]. B cell infection requires an additional glycoprotein, gp42, which binds to a B cell receptor, human leukocyte antigen (HLA) class II, through its C-type lectin domain (CTLD) [7,8]. gp42 is a unique glycoprotein to EBV and gp42 deleted virus can bind to B cells while it loses the ability to infect B cells, implying that gp42 is indispensable for B cell infection [2,9]. EBV shows a dual-tropic infection in which gp42 plays the role of tropism switch [10,11]. The binding of gp42 to gH/gL inhibits the interaction between gH/gL and the epithelial cell receptor, meaning that the virus carrying gp42 preferentially infects B cells [12]. Virions produced from B cells are lacking in gp42 as gp42 binds to HLA-II and is further degraded by protease in the endoplasmic reticulum (ER), which contributes to tropism for epithelial cells [13]. Hence, virions produced from B cells show tropism for epithelial cells. Virions produced from epithelial cells express gp42 and show tropism for B cells. The shuttles between epithelial cells and B cells may account for EBV persistence and transmission in infected hosts and even be the key factor for its evolution with humans over the last 60–80 million years [14,15]. gp42 also plays a significant role in immune evasion by interfering with HLA-II-restricted T cell immunity [16]. In the infected cells, gp42 binds to HLA-II-peptides immune complexes and impairs the recognition by T cell receptor (TCR) by creating steric hindrance [17].

gp42 is a type II membrane glycoprotein, with a flexible N-terminal domain binding to gH/gL and a CTLD binding to HLA-II [18,19,20]. The B cell entry triggering complex consists of gp42, gH/gL and HLA-II which is indispensable for B cell infection [21]. gp42 participates in membrane fusion by bridging HLA-II and gH/gL in an approximately parallel orientation, namely “closed” conformation, to bring the viral and host cell membranes into closer proximity [21]. gp42 binds to gH/gL with nanomolar affinity through its flexible N-terminal region, which has been mapping to residues 36-81 [20]. A recent study demonstrated that 33 residues consisting of aa 44-61 and aa 67-81 of gp42 are indispensable for maintaining the high affinity with gH/gL [22]. The mutations in the gp42 N-terminal domain disturbed the fusion activity, meaning that this region is essential for B cell fusion activation [20,22]. However, no monoclonal antibody (mAb) recognizing gp42 N-domain has been reported so far [23]. It is still unclear whether this region can induce neutralizing antibodies (nAbs) and become a targeted epitope for drug and vaccine development.

In this study, we have focused on the gp42 N-terminal region and generated a panel of mAbs targeting residues 44-61 and 67-81, respectively. These nine mAbs show high affinity with gp42 and can be used in an enzyme-linked immunosorbent assay (ELISA), flow cytometry, immunofluorescence (IF) and surface plasmon resonance (SPR), which are useful tools for investigating EBV infection mechanisms. The immunogenicity analysis demonstrates that the gp42 N-terminal region could induce sufficient neutralizing antibody titer to prevent EBV infection. This is the first systematic analysis of immunogenicity of the gp42 N-terminal region.

## 2. Materials and Methods

### 2.1. Construction, Expression and Purification

The gene of gp42 (Gene ID: 3783745) was synthesized and cloned into a pVRC8400 expression vector with an N-terminal CD5 signal peptide and a C-terminal 6xHis tag.

Wild-type and chimeric HBc149 particles were constructed as previously described [24]. Briefly, the sequences encoding gp42 (residues 44-61), (residues 67-81), (residues (44-61) × 3), (residues (47-81) × 3) and (residues 44-81) were cloned into pTO-T7-HBc149 between the restriction endonuclease sites of *BamH I* and *EcoR I*. The repeated sequences were connected with the (G_4_S)_2_ linker. All the above clones were confirmed by sequencing.

The sequences encoding gp42 (residues 44-61) and gp42 (residues 67-81) were inserted into pGEX-6p-1 (GST tag) between the restriction endonuclease sites of *BamH I* and *EcoR I*.

The biotinylated wild-type peptides and mutant peptides were synthesized by ChinaPeptides Co., LTD. All these peptides were characterized by HPLC-MS/MS and purity (not lower than 90%) was determined by HPLC-UV (220 nm).

Plasmid encoding gp42-His was transiently transfected into 293F cells using polyetherimide (PEI) with a mass ratio of 1:3 (Plasmids: PEI). The supernatant was collected 7 days after transfection. The cells and cell debris were removed by centrifugation at 12,500 rpm for 20 min. Then the supernatant was filtrated using a 0.22 µm filter. The gp42-His was purified with Ni^2+^ sepharose^TM^ 6 Fast Flow beads (GE Healthcare, Chicago, IL, USA). The target protein was eluted by elution buffer (250 mM imidazole in PBS, pH7.4). The purified protein was dialyzed into PBS overnight.

Plasmids of pTO-T7-HBc149-gp42(44-61aa), gp42(67-81aa), gp42(44-61aa) × 3, gp42(67-81aa) × 3, gp42(44-81aa) and wide-type HBc149 vector were transformed into BL21(DE3) competent bacteria. Protein production was induced by adding isopropyl-β-D-thiogalactoside (IPTG) at a final concentration of 0.2 mM when OD_600_ = 0.8 at 25 °C for 6–8 h. The bacteria were collected by centrifugating at 7000 g for 15 min and being resuspended by PBS. Then, the supernatant was collected after ultrasonication and centrifugation. The supernatant was heated at 65 °C for 30 min and centrifugated at 12,500 rpm for 10 min. The supernatant was further purified by gel filtration using Superdex 200 10/300 GL (Cytiva, Marlborough, MA, USA).

Plasmids of gp42(44-61aa)-GST and gp42(67-81aa)-GST were transformed into BL21(DE3) competent bacteria. Protein production was induced by adding isopropyl-β-D-thiogalactoside (IPTG) at a final concentration of 0.2 mM when OD_600_ = 0.8 at 25 °C for 6–8 h. The bacteria were collected by centrifugating at 7000 g for 15 min and being resuspended by PBS. Then, the supernatant was collected after ultrasonication and centrifugation. Next, the supernatant was filtered by 0.22 µm filter membrane. The proteins were further purified by gel filtration using GSTPrep FF 16/10 (Cytiva, Marlborough, MA, USA).

### 2.2. Immunization

For antibody screening, BALB/c mice (Shanghai SLAC Laboratory Animal Co., Ltd., Shanghai, China) were immunized subcutaneously three times at 2-week intervals with 100 µg gp42-His protein with complete Freund’s adjuvant and incomplete Freund’s adjuvant.

To evaluate the immunogenicity of the chimeric VLPs, BALB/c mice (*n* = 6 per group) (Shanghai SLAC Laboratory Animal Co., Ltd., Shanghai, China) and New Zealand white rabbits (*n* = 4 per group) (Songlian Laboratory Animal Center, Shanghai, China) were immunized three times at 2-week intervals with 20 µg (mice) or 100 µg (rabbits) corresponding proteins. Complete Freund’s adjuvant was used in the first injection and incomplete Freund’s adjuvant was used in the remaining two injections. The serum was collected before and after immunization at 2-week intervals for 8 weeks.

### 2.3. Antibody Screen and Preparation

After immunization, mice were boosted by administering soluble recombinant gp42-His. Three days later, the spleen cells were collected from immunized mice and fused with mouse myeloma Sp2/0 cells. The hybridomas were sequentially screened for the secretion of gp42-specific mAbs in the ELISA assay. The hybridomas were cloned five times and purified using AmMag^TM^ protein A magnetic beads (GenScript, Nanjing, China).

### 2.4. Enzyme Linked Immunosorbent Assay (ELISA)

The reactivity of the mAb with gp42-His was determined by indirect ELISA. In total, 200 ng/well of purified gp42-His in PBS was coated on 96-well ELISA plates at 37 °C for 2 h. The plate was washed by PBS containing 0.1% *v*/*v* Tween-20 (PBST) once and blocked with PBS containing 2% *w*/*v* non-fat dry milk (blocking solution) at 37 °C for 2 h. The 2-fold serial dilution of purified antibody was added to the wells and incubated at 37 °C for 30 min. After five washes with PBST, 100 µL of horseradish peroxidase (HRP)-conjugated goat anti-mouse IgG buffer was added to each well and incubated at 37 °C for 30 min. After five washes, 100 µL of tetramethylbenzidine (TMB) substrate (Beijing Wantai, Beijing, China) was added at 37 °C in the dark for 15 min. The reaction was stopped with a 2 M H_2_SO_4_ solution. Absorbance was measured at 450 nm using a PHOmo microplate reader (Autobio, Zhengzhou, China).

The antibody subtype was identified by goat anti-mouse IgG1, IgG2a, IgG2b, IgG3, IgM HRP (Abcam, MA, USA) using the ELISA assay.

To assess the binding ability of the immune serum, the serum samples were serially diluted 1:2 or 1:3 and applied to the protein-coated plate in 100 µL at 37 °C for 30 min. After five washes with PBST, 100 µL of HRP-conjugated goat anti-mouse or anti-rabbit IgG buffer was added to each well and incubated at 37 °C for 30 min. After five washes, 100 µL of TMB substrate was added at 37 °C in the dark for 15 min. The reaction was stopped with a 2 M H_2_SO_4_ solution. Absorbance was measured at 450 nm using a PHOmo microplate reader (Autobio, Zhengzhou, China).

The key residues recognized by each mAb were identified using ELISA. Streptavidin-coated plates (Beijing Wantai, Beijing, China) were coated with biotinylated peptides (200 ng/well) at 37 °C for 1 h. The 2-fold serial dilution of purified antibody was added to the wells and incubated at 37 °C for 30 min. After five washes with PBST, 100 µL of HRP-conjugated goat anti-mouse IgG buffer was added to each well and incubated at 37 °C for 30 min. After five washes, 100 µL of TMB substrate (Beijing Wantai, Beijing, China) was added at 37 °C in the dark for 15 min. The reaction was stopped with a 2 M H_2_SO_4_ solution. Absorbance was measured at 450 nm using a PHOmo microplate reader (Autobio, Zhengzhou, China).

### 2.5. Surface Plasmon Resonance (SPR) Assays

The equilibrium dissociation constant (KD values) for antibodies was performed using BIAcore 8k (Cytiva, Marlborough, MA, USA). The CM5 sensor was coated with gp42-His via covalent coupling. A 70 µL solution of a 1:1 (*v*/*v*) of N-ethyl-N-(3-diethylarninopropyl) carbodiimide (EDC) and N-hydroxysuccinimide (NHS) was mixed to activate the carboxyl groups on the dextran surface. The gp42-His was diluted with 10 mM sodium acetate (pH 5.5) to a final concentration of 5 µg/mL for coupling. In each channel, only flow cell 2 was coated with ligand, flow cell 1 was empty and blocked by ethanolamine as a control. After loading, the serially diluted antibody was injected at 30 µL/min for 300 s (association phase), followed by dissociating at 30 µL/min for 300 s (dissociation phase). The results were analyzed by BIAcore Insight Evaluation software, the curve fitting was performed using a 1:1 binding model.

### 2.6. Surface Staining of gp42 by Flow Cytometry

The plasmid of the full-length gp42 was transfected into 293T cells using PEI with a mass ratio of 1:3. The cells were trypsinized and resuspended by PBS and prepared by 10^6^ cells per test. The cells were stained with mAbs and negative control antibody 72A1 at 4 °C for 30 min. The cells were washed twice and incubated with goat anti-mouse IgG BV421 (Biolegend, San Diego, CA, USA) at 4 °C for 30 min in the dark. Stained cells were analyzed by flow cytometry on an LSRFortessaX-20 instrument (BD Biosciences, Franklin Lakes, NJ, USA) and the data were evaluated using FlowJo software X 10.0.7 (BD Biosciences, Franklin Lakes, NJ, USA).

### 2.7. Surface Staining of gp42 by Immunofluorescence

The 293β5 cells were transfected with full length EBV-gp42 and rhLCV-gp42 for 48 h. The cells were seeded on 96-well plates overnight. The cells were fixed with 4% paraformaldehyde at room temperature for 15 min and washed twice. The cells were incubated in a blocking solution (2% BSA in PBS) at 37 °C for 2 h and washed with PBST. Anti-gp42 mAbs were used for staining by incubation with cells at 4 °C overnight. After two washes with PBST, the cells were labelled with donkey anti-mouse IgG AlexaFluor 488 (Invitrogen, Carlsbad, California, USA) at room temperature for 30 min. After two washes with PBST, the cells were stained with DAPI (Invitrogen, Carlsbad, California, USA) for 5 min. The samples were visualized with the Opera Phenix Plus High-Content Screening System (PerkinElmer, Baesweiler, Germany).

### 2.8. SDS-PAGE and Western Blotting

The protein sample (loading buffer with SDS and β-mercaptoethanol) was subjected to 10% sodium dodecyl sulfate-polyacrylamide gel electrophoresis (SDS-PAGE) at 80 V for 1.5 h. Proteins were visualized by Coomassie brilliant blue staining for 30 min and destained until the background became transparent. Proteins on the gels were transferred onto polyvinylidene fluoride membranes (Millipore, CA, USA) after SDS-PAGE. The membranes were incubated with a blocking buffer (Beijing Wantai, Beijing, China) at room temperature for 2 h. The membranes were incubated with anti-gp42 mAbs, respectively. Next, 72A1 was used as a negative antibody. After five washes with PBST, the membrane was incubated with goat anti-mouse antibody conjugated with HRP (Abcam, Waltham, MA, USA) at room temperature for 30 min. The WesternBright ECL (Advansta, Menlo Park, California, USA) was used for color development.

### 2.9. Transmission Electron Microscopy (TEM)

The VLPs were analyzed by negative staining electron microscopy as previously reported [24]. Briefly, diluted unmodified HBc149 and chimeric HBc149 samples were applied to 200-mesh carbon-coated copper grids (Quantifoil, PA, USA) for 5 min and excessive solution were removed. After two washes with double distilled water, grids were immediately negatively stained for 30 s with freshly filtered 2% phosphotungstic acid (pH 6.4). Grids were examined with a FEI Tecnai T12 TEM (FEI, Portland, OR, USA) at an accelerating voltage of 120 kV and photographed at a magnification of 25,000-fold.

### 2.10. High Performance Size Exclusion Chromatography (HPSEC)

All unmodified HBc149 and chimeric HBc149 particles were analyzed by an 1120 Compact LC HPLC system (Agilent Technologies, Santa Clara, CA, USA) and separated by a TSK-Gel 5000PWxl 7.8 mm × 300 mm column (TOSOH, Minato-ku, Tokyo, Japan), which was pre-equilibrated in PBS. The flow rate and protein signal detection for the SEC analysis were 0.5 mL/min and 280 nm.

### 2.11. Homology Comparison

The alignment of EBV (Genbank ID: KF373730.1) and rhLCV (Genbank ID: AY037858) gp42 sequences was performed using ClustalW and ESPript 3.0.

The ClustalW website was used for sequence alignment and sequence alignment files were upload to the ESPript 3.0 website for drawing the sequence alignment result.

### 2.12. Virus Production

CNE2-EBV-GFP cells were induced by 12-O-tetradecanoylphorbol 13-acetate (TPA) (20 ng/mL) and sodium butyrate (2.5 mM) for 12 h and the media were replaced by RPMI1640 with 10% FBS. After 72 h, the cultures were clarified by centrifugation at 1000× *g* for 5 min and then the supernatant was filtrated through 0.45 µm filters. Viruses were concentrated 100 folds by centrifugation at 50,000× *g* for 2.5 h and the virus pellets were resuspended by RPMI1640 without FBS. The viruses were stored at −80 °C for use.

### 2.13. Neutralization Assay on B Cells

For B cell neutralization, the serums (2-fold dilution) collected from immunized mice and rabbits were mixed and incubated with 20 µL CNE2-EBV-GFP at 37 °C for 2 h. Subsequently, the mixture was added into 10^4^ cell/well Akata cells in 96-well plates. After being cultured in RPMI 1640 with 10% FBS for 48 h, the infection rate was determined by detecting the numbers of GFP positive cells using the LSRFortessaX-20 instrument (BD Biosciences, Franklin Lakes, NJ, USA). Uninfected cells were used as negative control and EBV infected Akata cells were used as positive control.

### 2.14. Statistical Analysis

All statistical analyses were performed with GraphPad Prism. *p*-values were generated by a one-way ANOVA analysis. Data were considered statistically significant at * *p* < 0.05.

## 3. Results

### 3.1. Generation and Characterization of gp42-44-61aa and gp42-67-81aa Specific mAbs

The gp42 N-terminal region is important for gH/gL binding (Figure 1A) while no mAb has been reported to recognize this domain. To obtain mAbs targeting the gp42 N-terminal region, recombinant EBV gp42 (aa 36-223) expressed by 293F cells was used for mice immunization and subsequent antibody screening. The hybridomas supernatants were screened for binding activity against gp42 protein, gp42-44-61aa and gp42-67-81aa. The positive clones reacting with protein and peptides (OD value ≥ 0.2) were chosen and progressed for subsequent screening rounds. Finally, six antibodies (2C3, 2E4, 3D3, 4D8, 6B8 and 6C1) specifically bound to gp42-44-61aa (Figure 1B) and three antibodies (4H7, 4H8 and 11G10) showed specificity to gp42-67-81aa (Figure 1C). We further determined the subtypes of these mAbs and found that most mAbs belonged to the IgG1 subclass and only 4H7 belonged to the IgG2a subclass (Figure 1D). We then evaluated the binding ability of gp42-specific mAbs by both ELISA and SPR assays. These mAbs showed great specificity toward gp42 with nanomolar affinity (Figure 1E and Appendix A, Table 1).

We further characterized the binding ability of mAbs to conformational epitopes using a flow cytometric and immunofluorescence (IF) analysis of 293T and 293β5 cells expressing full-length gp42. The results showed that all the mAbs had good reactivity with the native gp42 protein displayed on the cell surface while the negative control, 72A1, had no such binding activity (Figure 2A,B). A Western blot analysis showed that mAbs recognizing gp42-67-81aa had a good reactivity with denatured the gp42 protein and GST-fused gp42-67-81aa peptide (Figure 2C and Appendix A), suggesting that our mAbs recognized linear epitopes. Surprisingly, no mAbs to gp42-44-61aa reacted with the denatured gp42 protein or GST-fused peptide (Figure 2C and Appendix A), implying that the epitopes recognized by these antibodies showed resistance to denaturation.

### 3.2. Cross-Reactivity Analysis of mAbs Targeting gp42 N-Terminal Region

Animal models including nonhuman primates (NHP) are becoming important experimental systems for understanding EBV infection [25,26,27]. Rhesus macaques which can be naturally infected by rhesus lymphocryptovirus (rhLCV) have developed as an ideal surrogate for studying EBV infection [28,29,30,31]. It has been reported that rhesus macaques can reproduce the natural interaction between virus and host, leading to lifelong and latent infection with tumorigenic potential [25]. RhLCV encodes an identical repertoire of viral genes to EBV, with a remarkable homology [32,33]. The lytic-cycle proteins of EBV and rhLCV share 49–98% amino acid similarity and the latent-cycle proteins share 28–60% amino acid similarity [33]. A sequence analysis revealed that EBV and rhLCV gp42 sequences showed a high degree of conservation (79.64%) and their gp42-N terminal peptides shared 71.05% amino acid identity, implying the conserved function of gp42-44-81aa (Figure 3A and Appendix A). To assess antibody cross-reactivity, 293β5 cells transfected with rhLCV-gp42 were stained with our newly obtained mAbs and analyzed using an IF assay. The IF result demonstrated that only mAbs to gp42-67-81aa showed cross-reactivity while the remaining mAbs to gp42-44-61aa did not react with rhLCV gp42 (Figure 3B). We further confirmed the cross-reactivity of 4H7, 4H8 and 11G10 by indirect ELISA against synthesized rhLCV gp42-67-81aa peptides (Appendix A). Cross-reactive and not cross-reactive mAbs would be useful tools to identify the common and different epitope features between EBV and rhLCV gp42, aiding in better understanding the lymphocryptovirus infection mechanism.

### 3.3. Epitope Analysis of mAbs by Indirect ELISA

The antibody epitopes were directed to the gp42 N-terminal region whereas antibody activities were variable in binding affinity and cross-reactivity (Table 1, Figure 2 and Figure 3B), suggesting that these mAbs may bind to different core residues on the gp42 N-terminal region. We further sought to determine the key residues recognized by mAbs to explain the differences of the antibody binding activity. A panel of single alanine mutated peptides of gp42-44-61aa to gp42-68-81aa was used to detect the pivotal sites contributing to antibody binding. The amino acid in residue 67 of gp42 was naturally occurring alanine, which was regarded as a wild-type peptide (Figure 1A). mAbs recognized gp42-44-61aa targeted variable sets of key residues but most residues were located on gp42-53-56aa (Figure 4A and Appendix A). Surprisingly, the recognition of antigen by 2C3 was altered by most single alanine substitutions in gp42-44-61aa (Figure 4A). Furthermore, 2C3 could not react with the denatured gp42 protein (Figure 2B), indicating 2C3 may recognize discontinuous epitope or incomplete linear epitope, whose interaction may involve multiple key residues. The single alanine substitution of ^56^D abolished the binding activity of 4D8 (Figure 4A). Two mutated residues (W53A and V55A) significantly decreased 3D3 binding capacity (Figure 4A). Among the mutants, W53A altered the binding to gp42-44-61aa by 6B8 (Figure 4A). All the single mutations in gp42-44-61aa did not make a difference to the 2E4 and 6C1 binding ability, suggesting that the interaction between these two mAbs and gp42 was not determined by one residue alone. Alanine substitution results showed that 4H7 and 11G10 both recognized the key residue of ^81^W while a single amino acid mutation did not influence the 4H8 binding (Figure 4A). The key residues recognized by different mAbs were displayed in the structure model as shown in Figure 4B.

### 3.4. Construction and Characterization of Epitope-Displaying Chimeric VLPs

To further evaluate the function of the gp42 N-terminal region, we sought to assess the immunogenicity of this region. Firstly, we used bovine serum albumin (BSA) and keyhole limpet hemocyanin (KLH) as vectors to carry gp42-44-61aa and gp42-67-81aa peptides for immunization. The BSA and KLH conjugated gp42 N-terminal peptides could be specifically recognized by the anti-gp42 immune serum (Appendix A), indicating that the peptides were exposed on the surface of the carrier proteins. However, the fusion proteins could not induce a high-titer humoral response, especially for 67-81aa peptides (Appendix A), suggesting that a better peptide-displaying platform was needed. The truncated Hepatitis B virus core protein (HBc149) was a well-known virus-like-particle (VLP) and had been applied as an ideal epitope-displaying platform in EBV, HIV and EV71 vaccine research [24,34,35]. Hence, we also applied chimeric VLPs as a carrier protein to enhance the immunogenicity of peptides. Three linear sequences (gp42-44-61aa, gp42-67-81aa and gp42-44-81aa) derived from the gp42 N-terminal region were designed to be presented on a VLP surface (Figure 5A). To enhance the immunogenicity, gp42-44-61aa or gp42-67-81aa was repeated with three copies and connected with flexible linkers (Figure 5A). The soluble gp42 protein and unmodified HBc149 (wild-type) were used as controls in the subsequent evaluation. All the chimeric proteins were expressed in an *E. coli* system and further purified by column chromatography. The proteins showed clear bands in SDS-PAGE and mAb 1B11 against HBc149 showed specific binding in the Western blot analysis (Figure 5B). A negative-stained transmission electron microscope (TEM) and high-performance size exclusion chromatography (HPSEC) analysis demonstrated that all the proteins could spontaneously self-assemble into VLPs with similar sizes and morphologies compared to WT-HBc149 particles (Figure 5C). HBc149-specific mAb 1B11 and antiserum against gp42 efficiently bound to chimeric VLPs using the ELISA assay (Figure 5D). The results showed that all the inserted sequences did not influence the assembly of chimeric particles and the epitopes derived from the gp42 N-terminal region were successfully rebuilt and displayed on the particle surfaces. The homology modeling of the chimeric VLPs is displayed in Figure 6. However, the results were predicted and may not represent the actual structure of the chimeric VLPs.

### 3.5. Immunogenicity of the Chimeric VLPs

The immunogenicity of chimeric VLPs was evaluated using mice and rabbits (Figure 7A). The antibody titer against the gp42 protein was assessed using the ELISA assay. Chimeric VLPs induced a similar increase trend of antibody titers in mice and rabbits, which both reached peaks around 4.5-log to 5-log at week 6 immediately following the third injection (Figure 7B,C and Appendix A). Overall, the antibody titer elicited by chimeric VLPs in rabbits (5-log) was slightly higher than that of mice immunized with chimeric VLPs (4.5-Log) at week 8 (Figure 7D,E). On the contrary, soluble gp42 induced 1-log higher antibody titer in mice compared to that in rabbits from week 2 (Figure 7B,C). The results demonstrated that the immune response differed in different animal models when immunized with even identical antigen. Surprisingly, the repeated copies of gp42-44-61aa and gp42-67-81aa did not enhance the humoral immune response (Figure 7B,C). Similarly, gp42-44-81aa covering 44-61aa and 67-81aa did not help to increase the antibody titer. The multivalent single epitope displayed on the particle surface may already approach saturation points of antigen presentation and could not increase the antibody titer further.

We next sought to investigate the functional neutralizing activity of serum antibodies induced by gp42 N-terminal region-based chimeric VLPs. Considering that gp42 was only involved in the EBV infection of B cells, the B cell neutralization assay was used. Overall, the serum from chimeric VLPs immunized mice and rabbits showed blocking activity against EBV infection (Figure 7F,G and Appendix A). The immunization of the intact extracellular domain of gp42 stimulated a potent neutralizing humoral response (Figure 7F,G). Consistent with the binding activity result, the repeated copies of single peptides did not significantly promote the neutralizing activity of the immune serum (Figure 7F,G). Similarly, the humoral response against identical antigen varied in mice and rabbits (Figure 7F,G). In summary, chimeric VLPs carrying gp42 N-terminal peptides could induce a gp42-specific humoral immune response, which neutralized the EBV infection of B cells.

## 4. Discussion

gp42 establishes an important connection between virus and host cells by binding to gH/gL on the viral membrane through the N-terminal region and interacting with HLA-II on the cellular membrane through CTLD [36,37,38]. Few mAbs targeting gp42, especially targeting the gp42 N-region, have been reported and little research has been conducted to assess the immunogenicity of the gp42 N-terminal region.

In this study, a panel of nine mAbs against gp42 was generated by immunizing BALB/c mice with recombinant gp42 proteins and we have comprehensively characterized them. We reported antibodies recognizing the gp42 N-terminal region and analyzed the activity of antibodies for the first time. The ELISA and SPR results showed that nine mAbs had specific binding ability and good affinity (Figure 1E and Table 1). The flow cytometry and IF results demonstrated that all mAbs recognized gp42 expressed on the cell membrane surface (Figure 2), indicating that these mAbs could well recognize the native form of gp42. These findings suggested that all nine mAbs could be widely used in detecting and labeling recombinant and native gp42 proteins in vivo and in vitro.

RhLCV, the rhesus viral homolog of EBV, can naturally infect rhesus macaques and reproduce the process of infection. Rhesus macaques are considered to be the suitable animal model for EBV infection and widely used in EBV infection mechanisms, neutralizing antibodies assessment and vaccines development [39,40,41]. RhLCV shows genomic similarity with EBV, while limited numbers of mAbs cross-reacted with both EBV and rhLCV [39,41] and no gp42 mAbs showed cross-reactivity so far. Hence, extra efforts were needed to construct a chimeric virus by replacing genes on rhLCV with corresponding genes of EBV for antibody or vaccine evaluation. In the present study, we reported that mAbs (4H7, 4H8 and 11G10) targeting gp42 showed cross-reactivity with rhLCV gp42 for the first time (Figure 3B). EBV and rhLCV gp42 share 79.64% amino acid identity while not all the nine mAbs reacted with rhLCV gp42 (Figure 3B). The differences of mAbs in cross-reactivity may be attributed to epitope diversity (Figure 4A,B). The gp42 antibody panel may be a useful tool for investigating the gp42 function both in EBV and rhLCV infection mechanisms and why identical epitopes could induce antibodies with different activities including cross-reactivity. In summary, the newly obtained gp42 mAbs provided a useful tool to study the infection mechanism and conserved feature of EBV and rhLCV.

Up to now, only several anti-gp42 neutralizing antibodies including F-2-1 have been reported [42]. MAb F-2-1 efficiently inhibited B cell infection, which recognized the region involved in HLA-II binding and exhibited neutralizing ability by blocking the gp42 and HLA-II interaction [13,43,44]. We generated nine mAbs whose epitopes were mapped to the gp42 N-terminal region and unfortunately no mAb showed obvious neutralizing activity in the B cell infection model (data not shown). gp42 is divided into two main regions: N-terminal region and CTLD. CTLD is essential for receptor binding and therefore more likely to induce antibodies with the receptor blocking ability. It is common sense that receptor binding domains of viral glycoproteins are always considered as major targets to elicit neutralizing antibodies. This may explain why we cannot obtain functional antibodies targeting the gp42 N-terminal region which has no direct connection to receptor binding. However, it is a fact that the number of our antibodies is not enough to draw an exact conclusion. Further antibody screening targeting gp42 N-terminal region is needed to answer the question of whether or not the gp42 N-terminal region can induce neutralizing antibodies.

We further evaluated the immunogenicity of gp42 N-terminal peptides. Peptides conjugated to BSA and KLH were used for immunization at first. However, BSA and KLH conjugated gp42-44-61aa peptides could only induce low levels of antibody titer against gp42 protein. In addition, BSA and KLH conjugated gp42-67-81aa peptides elicited a lower antibody titer (Appendix A), suggesting that BSA and KLH were not the ideal choice for carrier proteins. In order to promote the immunogenicity of these peptides, we constructed chimeric VLPs carrying the gp42 N-terminal peptides (Figure 5A). Unlike the BSA and KLH conjugated peptides, chimeric VLPs significantly increased the immune serum titer (Figure 7A,B). The slight differences of immunogenicity of the identical chimeric VLP in mice and rabbits revealed that the humoral immune response varied in different animal models, which should be taken into consideration during the evaluation of EBV vaccines. We also found that the repeated copies of a single epitope at the monomer of the particle did not induce a higher antibody titer. A suitable range among antigenic epitopes is needed for B cell receptor cross-linking and microclustering [45,46]. The direct repetition of one peptide in a display unit may be too dense to activate B cells. Surprisingly, the chimeric VLP-immunized serum did have a functional neutralizing activity while the gp42 protein generated remarkably higher antibody titers with strong EBV infection blocking efficiency (Figure 7F,G). The chimeric VLPs were expressed using the prokaryotic expression system, which may have the defects of lacking a correct conformational structure and post-translational modification [47,48]. There were four N-linked glycosylation sites on the gp42 protein, one of which was located at ^64^NKT^66^ and did not involve either region of gp42-44-61aa or gp42-67-81aa [13]. The missing of glycosylation may also impair the epitope reconstruction. Our chimeric VLPs could be specifically recognized by the gp42-immunized serum, while these foreign peptides could only be exposed in a more linear fashion, and were unlikely to induce conformational-dependent antibodies. To date, most anti-EBV neutralizing antibodies are conformation-dependent, suggesting conformational epitopes are indispensable for inducing a neutralizing antibody response. Besides, the instability (protease-sensitivity) of exposed peptides on the surface of chimeric VLPs may also interpret lower neutralization activity exhibited by the peptide constructs. Hence, more rational designs of chimeric VLPs and the eukaryotic expression system are required to achieve a better functional humoral response. On the other hand, gp42 binds to gH/gL with high affinity, forming a stable heterotrimer on the mature EBV virion membrane. The formation of gH/gL/gp42 heterotrimer inhibited or competed with antibodies induced by gp42 N-terminal peptides binding to gp42. This might explain why the neutralization results of gp42 N-terminal peptides were less potent than those of the soluble gp42 protein which contained the HLA-II binding site (neutralization site).

In this study, we generated and characterized a panel of nine mAbs to the gp42 N-terminal region. These mAbs showed high specificity against recombinant and native gp42 proteins and variable cross-reactivity against rhLCV gp42. These mAbs would be useful tools in investigating the lymphocryptovirus infection mechanism. We also carried out a systematic analysis of the immunogenicity of the gp42 N-terminal region for the first time. More efforts are needed to design a better vaccine based on gp42 N-terminal peptides.

## Figures and Tables

**Figure 1 viruses-13-02380-f001:**
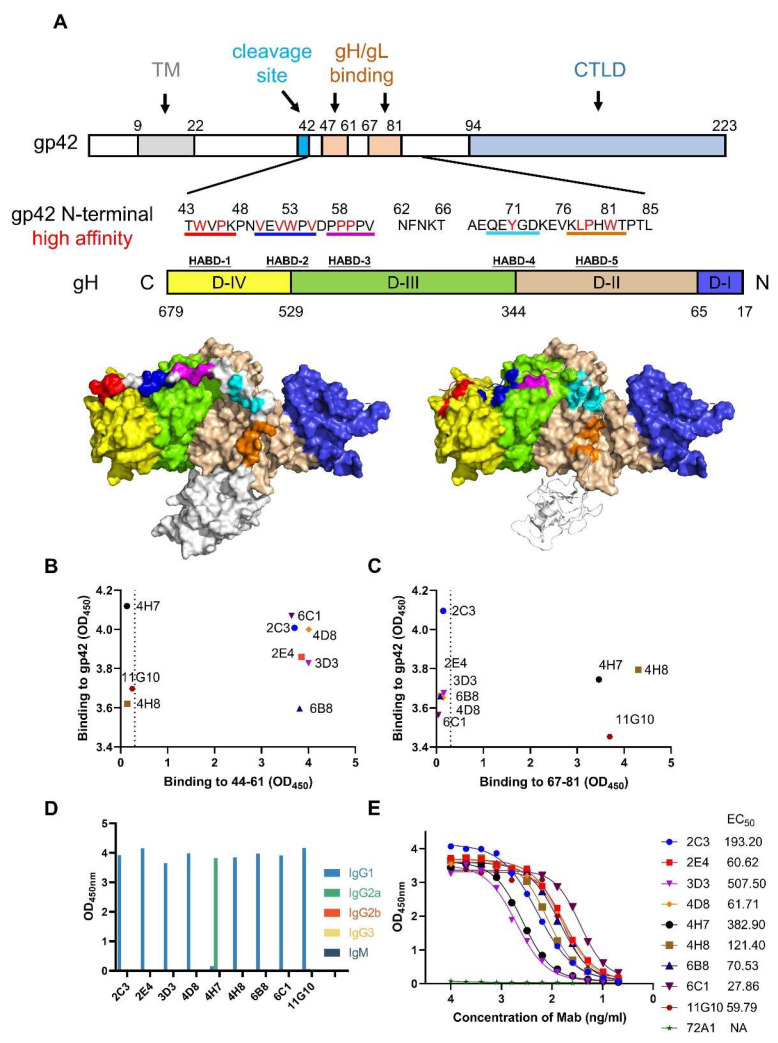
Generation of gp42 N-terminal region-specific mAbs. (**A**) Graphical representation of gp42 and gH/gL protein. The major functional domains of gp42 are indicated by labeling with different colors. Transmembrane domain (TM) in gray, cleavage site in cycan, gH/gL binding site in light orange and C-terminal lectin domain (CTLD) in light blue. The amino acids of gp42 N-terminal region are listed and residues interacting with gH are colored in red. gH domains (D-IV to D-I) are colored in yellow, green, light brown and blue, respectively. Structure models of gH/gL/gp42 show the high-affinity binding determinants (HABDs) of gH and corresponding interacting regions on gp42, which are colored in red, blue, magenta and orange, respectively. Structure models were drawn with PyMOL and the PDB accession number of gH/gL/gp42 is 5T1D. (**B**,**C**) The binding ability of hybridomas to gp42 protein and (**B**) gp42-44-61aa peptide or (**C**) gp42-67-81aa peptide were assessed using ELISA assay. (**D**) Identification of antibody subtypes. (**E**) Binding activities of antibodies were tested and the EC_50_ values were calculated.

**Figure 2 viruses-13-02380-f002:**
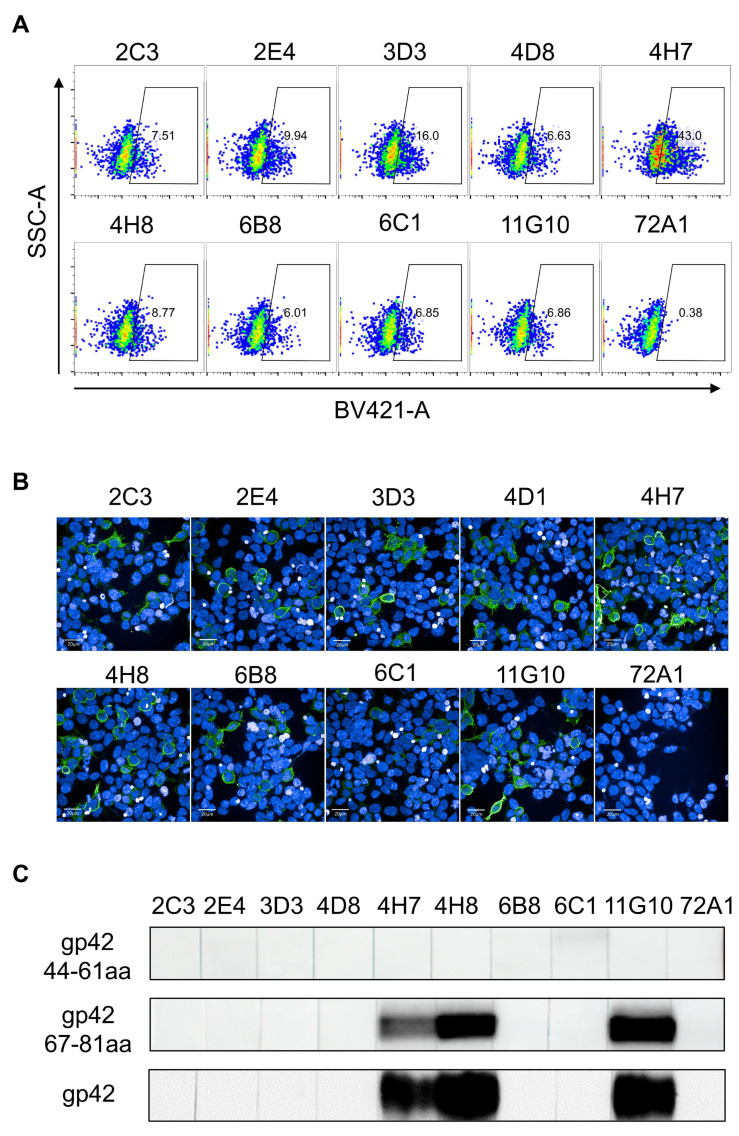
Evaluation of binding activities of 9 mAbs. (**A**,**B**) Detection of gp42 on the surface of full-length gp42 transfected 293T cells using (**A**) fluorescence activated cell sorting (FACS) and (**B**) immunofluorescence (IF) assays. Antibodies were labeled with (**A**) GAM-BV421 and (**B**) GAM-Alexa-Fluor 488. Anti-gp350 murine antibody 72A1 was used as negative control. (**C**) In total, 9 mAbs were tested by Western blot for reactivity against denatured GST-fused gp42-44-61aa, GST-fused gp42-67-81aa and gp42 proteins.

**Figure 3 viruses-13-02380-f003:**
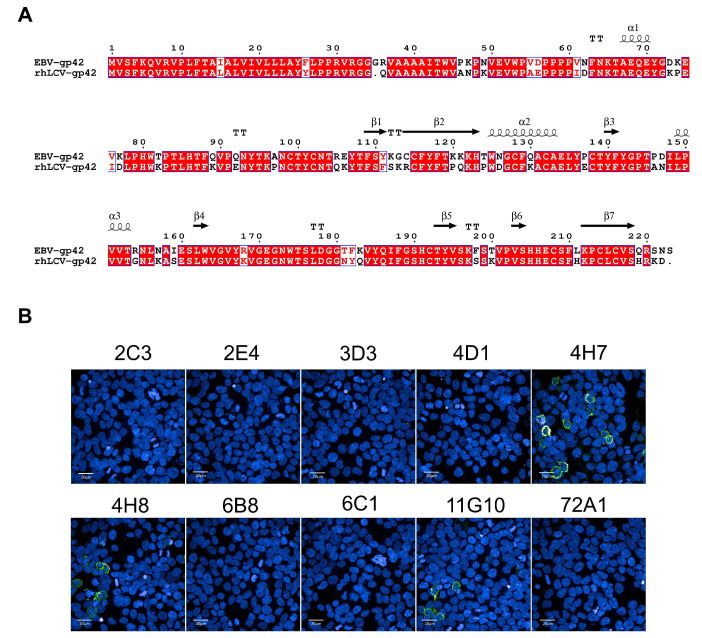
The cross-reactivity with rhLCV-gp42 of 9 mAbs. (**A**) Alignment of EBV (Genbank ID: KF373730.1) and rhLCV (Genbank ID: AY037858) gp42 using ESPript 3.0. The identical amino acids are marked in the red color. (**B**) Detection of the binding abilities of 9 mAbs with rhLCV-gp42 transfected 293β5 cells. The 72A1 was used as negative control.

**Figure 4 viruses-13-02380-f004:**
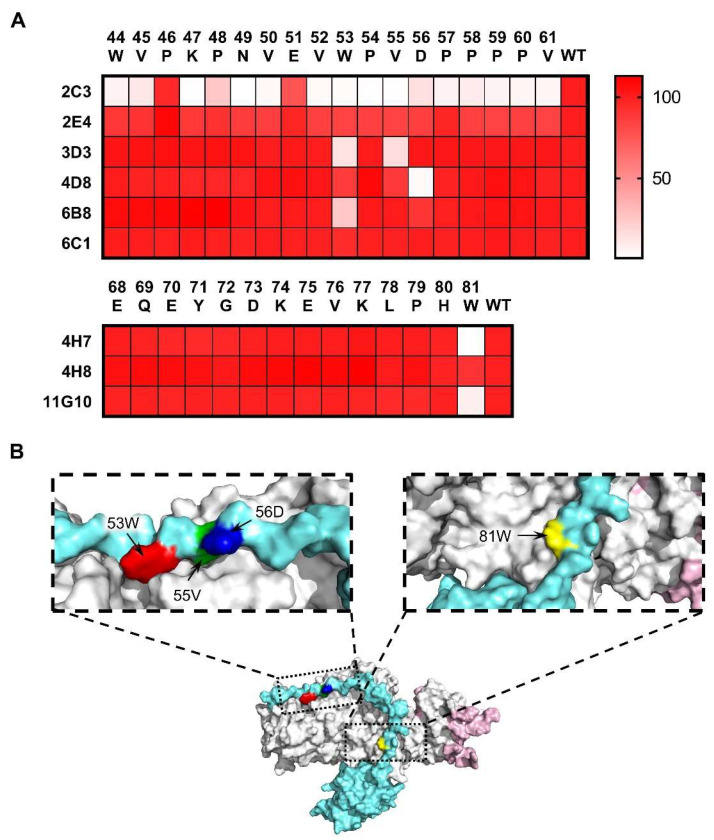
Key residue identification of anti-gp42 mAbs. (**A**) Alanine scanning libraries (gp42-44-61aa and gp42-67-81aa) were used to identify the key residues involved in gp42 and antibody interaction. Relative binding potency was calculated in percentage terms. (**B**) The key residues recognized by anti-gp42 mAbs are displayed. Amino acids (^53^W, ^55^V, ^56^D and ^81^W) are labeled in red, green, blue and yellow, respectively. Images were edited using PyMOL and the PDB accession number is 5T1D.

**Figure 5 viruses-13-02380-f005:**
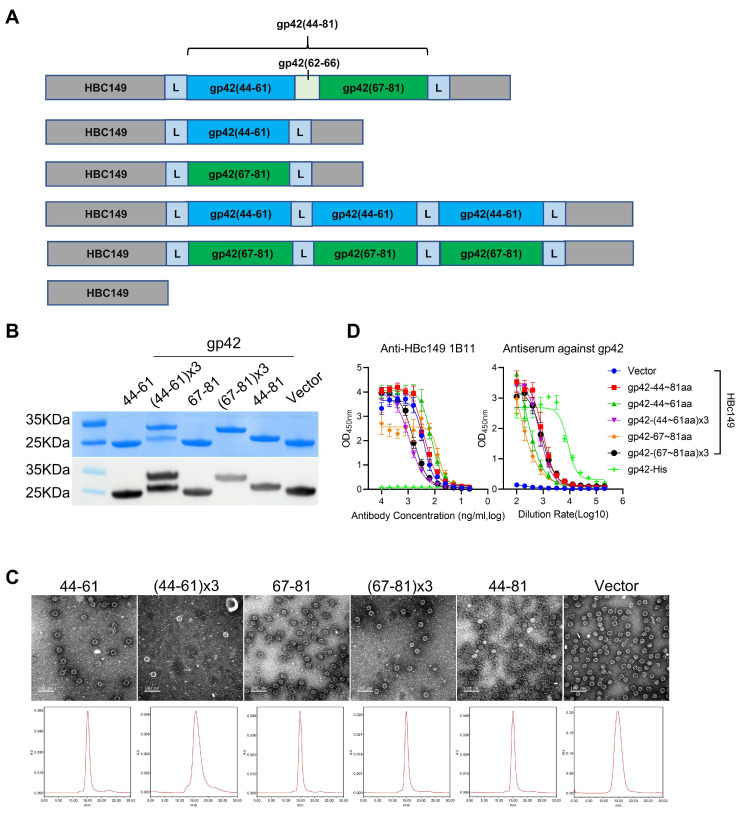
Generation and characterization of chimeric VLPs. (**A**) Schematic diagram illustrating the design of chimeric VLPs carrying the peptides from gp42 N-terminal region. gp42-44-61aa is labelled in cyan and gp42-67-81aa is labelled in green. (**B**) SDS-PAGE and Western blot analysis of fusion proteins. Anti-HBc149 mAb 1B11 was used for particle detection. (**C**) The size and morphology analysis evaluated by transmission electron microscope (TEM) and high-performance liquid chromatography (HPSEC). (**D**) Reactivities of chimeric VLPs with anti-gp42 immune serum and anti-HBc149 mAb 1B11 using ELISA assay.

**Figure 6 viruses-13-02380-f006:**
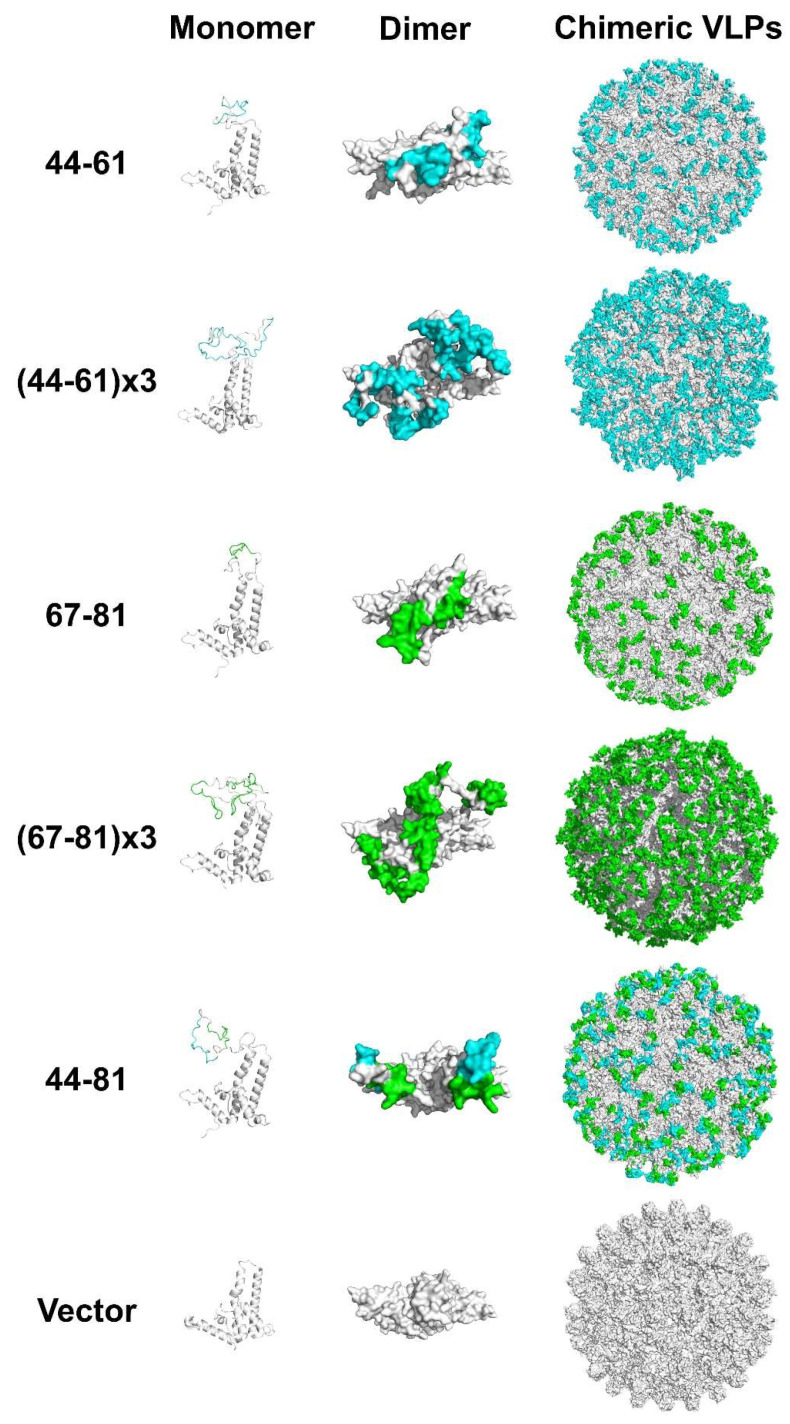
Structure models of chimeric VLPs. The monomer form of fusion proteins was drawn in cartoon patterns and listed in the left. The dimer form of fusion proteins was edited in surface patterns and placed in the middle. Chimeric VLPs were rendered in surface patterns and set in the right. The fusion proteins were the homology modeling result using SWISS-MODEL (the template PDB ID: 3J2V).

**Figure 7 viruses-13-02380-f007:**
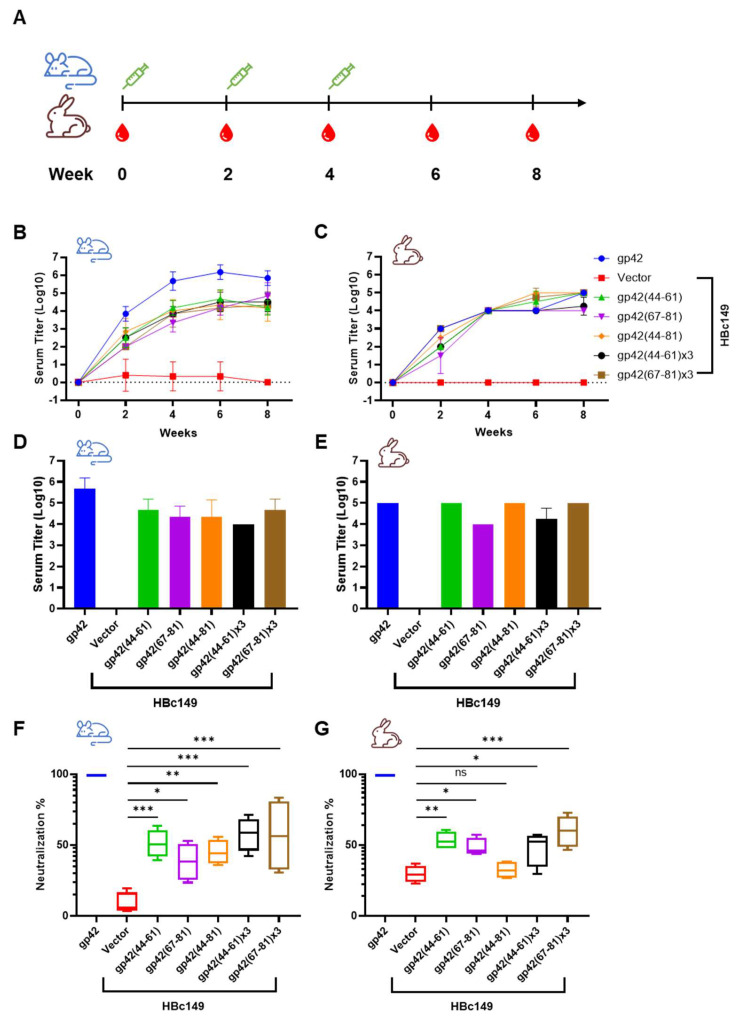
Immunogenicity analysis of chimeric VLPs. (**A**) Schematic diagram of immunization procedure. Mice (*n* = 6) and rabbits (*n* = 4) were immunized three times at 2-week intervals. Blood samples were collected five times at 2-week intervals. (**B**,**C**) The serum titers of (**B**) mice and (**C**) rabbits against gp42 from different timepoints were assessed by ELISA assay. (**D**,**E**) The antibody titers of (**D**) mice and (**E**) rabbits against gp42 were evaluated at week 8. (**F**,**G**) Neutralizing assessment of serum samples collected at week 8 from (**F**) mice and (**G**) rabbits. All data are presented as mean ± SEM. * *p* < 0.05; ** *p* < 0.01; *** *p* < 0.001; ns, not significant.

**Table 1 viruses-13-02380-t001:** The binding affinity of nine mAbs to gp42 were measured by SPR.

mAbs	Ka (10^5^/Ms)	Kd (10^−4^/s)	KD (nM)
2C3	0.79	10.90	13.80
2E4	1.35	2.47	1.82
3D3	0.47	9.70	20.70
4D8	1.52	5.99	3.93
4H7	0.36	0.62	1.73
4H8	0.30	1.27	4.32
6B8	2.16	5.86	2.72
6C1	1.24	3.60	2.90
11G10	0.36	0.33	0.93

## Data Availability

Not applicable.

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
