# Peer review of "Antibody Generation and Immunogenicity Analysis of EBV gp42 N-Terminal Region"

_viruses, 2021, doi:10.3390/v13122380_

Round 1

Reviewer 1 Report

The manuscript contains a number of grammatical errors, misuse of words and verb tenses or non-normal phrases in sentence structures. The paper would be improved if re-edited by an English editor. One may consider one offered by MDPI.

Example given below are a representative sample and not a complete edit.

line 39 precess

Line 50 On the contrary

line 309 hehaved

line 329 lied

line 454 reproted

Major comment:

The peptides used for immunization reflect the core gH/gL binding region of gp42 and encompass residues 47 to 56 and 67 to 81. One would assume that mature EBV GFP-virus gp42, through the peptides in question are in complex with gH and gL and not accessible to  antibody. This would explain why the neutralization results were less than expected as compared to full-length gp42 which contained the dominant HLA-II interaction/neutralization site. The alternative explanation outlined above should be discussed.

Minor comments:

One should clearly indicate that the VLP computer generated models for chimeric proteins may not represent the actual structure of the HBc149-gp42 peptide fusion proteins.

In addition to conformational changes, one may also interpret lessor neutralization activity exhibited by the peptide constructs due to peptide [protease] stability as compared to the stability of native gp42 following mouse or rabbit injection.

As a note the authors the immunization protocol is aggressive at two weeks and better results may occur if the animals were given an additional week to recover.

Author Response

Point 1: The manuscript contains a number of grammatical errors, misuse of words and verb tenses or non-normal phrases in sentence structures. The paper would be improved if re-edited by an English editor. One may consider one offered by MDPI.

Response 1: Thank you for your significant reminding. We are sorry for the mistakes in this manuscript and inconvenience they caused in your reading. According to your suggestion, we have corrected the above grammatical errors and polished the whole manuscript carefully. We believed that the revised manuscript is understandable to readers.

Point 2: The peptides used for immunization reflect the core gH/gL binding region of gp42 and encompass residues 47 to 56 and 67 to 81. One would assume that mature EBV GFP-virus gp42, through the peptides in question are in complex with gH and gL and not accessible to antibody. This would explain why the neutralization results were less than expected as compared to full-length gp42 which contained the dominant HLA-II interaction/neutralization site. The alternative explanation outlined above should be discussed.

Response 2: We appreciate for your valuable comment. We have added the alternative explanation in line 508-513 in the revised manuscript.

“On the other hand, gp42 binds to gH/gL with high affinity, forming a stable heterotrimer on the mature EBV virion membrane. The formation of gH/gL/gp42 heterotrimer inhibited or competed with antibodies induced by gp42 N-terminal peptides binding to gp42. This might explain why the neutralization results of gp42 N-terminal peptides were less potent than that of soluble gp42 protein which contained the HLA-II binding site (neutralization site).”

Point 3: One should clearly indicate that the VLP computer generated models for chimeric proteins may not represent the actual structure of the HBc149-gp42 peptide fusion proteins.

Response 3: Thanks for your nice suggestion. We have added a brief description as follows: “The homology modeling of the chimeric VLPs was displayed in Figure 6. However, the results were predicted and may not represent the actual structure of the chimeric VLPs.” in line 378-380 in the revised manuscript.

Point 4: In addition to conformational changes, one may also interpret lessor neutralization activity exhibited by the peptide constructs due to peptide [protease] stability as compared to the stability of native gp42 following mouse or rabbit injection.

Response 4: Thank you very much for your kindly comment. We have added a brief description as follows: “Besides, the unstability (protease-sensitivity) of exposed peptides on the surface of chimeric VLPs may also interpret lower neutralization activity exhibited by the peptide constructs.” in line 504-506 in the revised manuscript.

Point 5: As a note the authors the immunization protocol is aggressive at two weeks and better results may occur if the animals were given an additional week to recover.

Response 5: Thanks for your suggestion. We found that the immunized serum titer showed increasing trend using current immunization protocol. We would consider your kindly suggestion carefully and prolong the immunization interval in our further work.

Reviewer 2 Report

The manuscript describes the design, construction and generation of monoclonal antibodies against the N-terminal region of gp42 of EBV. the authors generated 9 mAbs where 6 of them are specifically interacting with EBV gp42 and 3 are cross reacting with ehLCV gp42. The authors confirmed the reactivity with different assays.

The manuscript is well written and properly describes all steps and results.

only minor comment where the authors forgot to delete the originally written template sentences in different places (lines 80-88; 446-448; 537-540).

Author Response

Point: only minor comment where the authors forgot to delete the originally written template sentences in different places (lines 80-88; 446-448; 537-540).

Response: Thanks for your suggestion. We have removed template sentences in the latest version of our manuscript.

Reviewer 3 Report

EBV is one of the most common human viruses in the world. However, despite the global health burden that EBV poses, an effective EBV prophylactic vaccine remains elusive. EBV glycoproteins complex gH/gL along with gp42 is indispensable for virus entry in B-cells and pg42 also play role in immune evasion in virus-infected cells. However, no antibodies against the N-terminal region of gp42 are available and the immunogenicity of the region is largely unknown. 

Here the authors have generated a panel of monoclonal antibodies (mAb) targeting the N-terminal region of gp42 protein of Epstein-Barr virus (EBV) and demonstrated that gp42 N-terminal region as virus-like particle could induce sufficient neutralizing antibody titer to prevent EBV infection in B-cells. They have generated a total of 9 mAb (6 mAbs to 44-61 amino acid residues and 3 mAbs to 67-81 amino acid residues of gp42) and among them, 3 mAbs cross-reacted with gp42 of rhesus lymphocryptovirus (rhLCV) closely related to EBV. These newly produced mAbs and immunogenicity studies could help in further investigation of EBV infection mechanism and rational design of effective vaccines.

Minor critiques and suggestions:

Figure 2A and B: inclusion of a gp42 antibody-like F21 would have been a good positive control to show that it binds to full-length gp42 as well.

Figure 3A: It would be great if the authors show where the newly developed antibodies bind in the sequence.

Figure 7: For panel B-E, indicating which graph belongs to mice vs rabbit would help with quick identification.

Figure 7: panel F and G, the sera from all the rabbits and mice immunized with gp42 recombinant protein, showed complete neutralization?

Author Response

Point 1: Figure 2A and B: inclusion of a gp42 antibody-like F21 would have been a good positive control to show that it binds to full-length gp42 as well.

Response 1: Thanks for your suggestion. Indeed, it would be good if mAb F-2-1 is used as a control here. However, purified antibody and the hybridoma cell line of mAb F-2-1 are not commercially available. Besides, the coding sequences of mAb F-2-1 light and heavy variable region are not published, resulting in a failure to express the mAb F-2-1. This work will be considered in our future work.

Point 2: Figure 3A: It would be great if the authors show where the newly developed antibodies bind in the sequence.

Response 2: Thank the reviewer for the comments. We have mapped the newly developed antibodies to the N-terminal region of gp42. The epitopes of these mAbs have been defined. And these mAbs could be divided into two groups whose epitopes could be located on residues 44-61 and residues 67-81 of gp42 respectively. The length of these two sequences were the length of B-cell epitopes. We believe our provided data are informative enough to understand the function of gp42 N-terminal region. 

Point 3: Figure 7: For panel B-E, indicating which graph belongs to mice vs rabbit would help with quick identification.

Response 3: Thank you so much for your nice suggestion. We feel sorry for the inconvenience brought to the reviewer. We have re-drawn Figure 7 according to the reviewer’s suggestion in line 423.

 Point 4: Figure 7: panel F and G, the sera from all the rabbits and mice immunized with gp42 recombinant protein, showed complete neutralization?

Response 4: Thank you. gp42 is the indispensable component and plays an important role in receptor binding during EBV infection in B cells. Previous studies showed that immunogens including gp42 recombinant protein showed potent neutralization (See references below). We have added the data of the ratio of infected cells from different immunized groups in Figure S9 in supplementary material. The data from Figure 7F, 7G and S9 showed that the sera from all the rabbits and mice immunized with gp42 recombinant protein showed complete neutralization.

References:

1. Immunization with Components of the Viral Fusion Apparatus Elicits Antibodies That Neutralize Epstein-Barr Virus in B Cells and Epithelial Cells.

2. A Pentavalent Epstein-Barr Virus-Like Particle Vaccine Elicits High Titers of Neutralizing Antibodies Against Epstein-Barr Virus Infection in Immunized Rabbits.

3. https://investors.modernatx.com/news-releases/news-release-details/moderna-reports-third-quarter-fiscal-year-2021-financial-results

Round 2

Reviewer 1 Report

The authors have vastly improved the quality of the paper.